# Experimental Study on the Reciprocating Shear Characteristics and Strength Deterioration of Argillaceous Siltstone Rockfill Materials

**Jun Du [1],\*** , **Dong Li [2]**, **Zhiming Xiong [3]**, **Xinggang Shen [1]**, **Chenchen Li [1]** and **Weiwei Zhu [1]**

[1] College of Architecture and Civil Engineering, Kunming University, Kunming 650214, China; kmu_shenxg@126.com (X.S.); wendyli123@163.com (C.L.); zwwswfc@126.com (W.Z.)
[2] Faculty of Land Resource Engineering, Kunming University of Science and Technology, Kunming 650093, China; lidong990201@126.com
[3] Faculty of Public Safety and Emergency Management, Kunming University of Science and Technology, Kunming 650093, China; xzm_0926@126.com
\* Correspondence: dujun0605@126.com

**Abstract:** The reciprocating shear mechanical properties and strength deterioration mechanisms of rockfill materials are of great research significance for high-fill slope stability analysis. To study the shear strength characteristics of argillaceous siltstone rockfill materials with different fabric characteristics under reciprocating shear loading, we analyzed the shear strength, hysteresis loop area, damping ratio, shear strength parameter, and shear stiffness of coarse-grained soils with different coarse grain contents using a coarse-grained soil direct shear testing machine capable of reciprocating shear and revealed their strength deterioration mechanism. The test results show that the shear strength of argillaceous siltstone rockfill materials is significantly affected by the coarse grain content and the number of reciprocating shears. Specifically, the shear strength increases with the coarse grain content and decreases with the number of reciprocating shears. The hysteresis loop area is positively correlated with the coarse grain content and negatively correlated with the number of reciprocating shears. The damping ratio is not related to the coarse grain content but tends to decrease with the number of reciprocating shears. Soil cohesion and the internal friction angle increase with the coarse grain content and decrease with the number of reciprocating shears. The soil failure shear stiffness is linearly correlated with the coarse grain content, and the normalized shear stiffness is logarithmically related to the number of reciprocating shears. According to these relationships, an empirical formula for the shear stiffness of argillaceous siltstone rockfill materials under different coarse grain contents and different numbers of reciprocating shears can be established to provide a basis for analyzing rockfill stability.

**Keywords:** rockfill material; coarse grain content; reciprocating shear; shear strength; strength deterioration

## 1. Introduction

Argillaceous siltstone rockfill materials are a type of loose geologic body stripped and piled up during the open-pit mining of phosphate ores. Under gravity sorting, the particle size distribution of the rockfill material shows significant grading characteristics at different heights of the dump [1]. The percentages of coarse and fine grains vary, and soil fabric characteristics differ, making the coarse grain content an important factor affecting soil strength characteristics [2–4]. Unlike continuous materials, rockfill materials are a typical class of discrete granular materials that can withstand reciprocating shear deformation in extreme geological environments [5,6], and studying their shear strength characteristics and strength deterioration mechanisms is of great research significance for high-fill slope stability analysis.

It has been shown that the coarse grain content significantly affects the strength and deformation properties of coarse-grained soils [7–10]. Chang et al. [11] found that grain

grading significantly affected the overall strength of coarse-grained soils and that increasing the gravel content decreased their strength compared to pure sand. Xu et al. [12] pointed out that the structural characteristics between soil grains changed with the increasing coarse grain content, which in turn affected the soil shear strength. Cui et al. [13] showed that mixed soils were significantly influenced by their coarse grain contents, and their shear strength properties showed different trends in different coarse grain content intervals. Rezvani et al. [14] tested calcareous sand and found that particle size distribution affected the stress–strain characteristics of soil grains. Zhang et al. [15] showed that the shear strength of coarse-grained soils increased by 82% to 174% as the coarse grain content increased from 30% to 60%. Chen et al. [16] pointed out that the internal friction angle and maximum shear dilatancy of red sandstone soils increased gradually with the increase of the coarse grain content, while the cohesion and maximum shear contraction increased and then decreased. Li et al. [17] conducted shear tests on sliding zone soils with different coarse grain contents, which exhibited significant shear dilatant characteristics at high coarse grain contents.

Disturbed by loads, soils as discrete grain aggregations with a pile structure undergo multiple shear deformations [18–20]. The stress–strain variation is fundamental for analyzing soil reciprocating shear characteristics. Asadzadeh et al. [21] found that with reciprocating shears, the uneven distribution of volumetric strain within coarse-grained soils was an important cause of local dilatancy and contraction behaviors. Saberi et al. [22] stated that soil stress hardening and stress softening were caused by the mechanical interaction between soil grains under reciprocating shear. He et al. [23] pointed out that the dynamic stress curves of coarse-grained soils under traffic loads varied nonlinearly and showed attenuation characteristics. Wang et al. [24] pointed out that red clay exhibited softening characteristics during reciprocating shear, and the shear stress decreased with the number of reciprocating shears. Liang et al. [25] pointed out that reciprocating shear behavior was the main cause of stress attenuation in mudstone grains. To investigate soil reciprocating shear characteristics, it is also necessary to analyze soil mechanical property indexes, including the shear strength, shear modulus, shear stiffness, hysteresis loop area, and damping ratio. Vieira et al. [26] found that the damping ratio at the sand–geotextile interface tended to increase under reciprocating shear. Wang et al. [27] stated that the damping ratio at the soil–geogrid interface decreased with the number of reciprocating shears. Zhang et al. [28] pointed out that the damping ratio of frozen soils with different coarse grain contents tended to decrease under cyclic loading. Cen et al. [29] found that the shear stiffness at the geomembrane–sand interface was significantly affected by the vertical pressure during reciprocating shear, which increased with the increase of the vertical pressure. Fakharian et al. [30] found that the shear modulus of mortar mixtures decreased under reciprocating shear while the damping ratio tended to increase. Wu et al. [31] found that as the number of reciprocating shears increased, the hysteresis loop area of mud–sandstone mixtures decreased while the damping ratio increased.

Previous studies have identified coarse grain content as an important factor affecting soil fabric characteristics and shear strength, and the soil load-bearing characteristics deteriorate with reciprocating shears. However, for rockfill materials with disparate grain sizes and significant grading in dumping fields, their strength deterioration mechanism under reciprocating loads is still unclear and needs in-depth studies. Therefore, a coarse-grained soil testing machine capable of reciprocating shear was employed to conduct push–pull shear tests on argillaceous siltstone rockfill materials with different coarse grain contents from a phosphate mine dumping field. We analyzed the variations of the soil shear strength, hysteresis loop area, damping ratio, shear strength parameter, and shear stiffness with the number of reciprocating shears and revealed the soil strength deterioration mechanism. The research results are expected to provide useful guidance for slope stability analysis in dumping fields.

## 2. Rockfill Material Reciprocating Shear Test

### 2.1. Test Equipment

Figure 1 shows the DHJ-30 coarse-grained soil shear testing machine (Figure 1a) used in the tests and its structural schematics (Figure 1b). The testing machine is equipped with a computerized automatic data acquisition and processing system, capable of collecting and storing load and displacement data and the automatic control of the vertical load, horizontal load, shear speed, etc. The shear box of the testing machine is $\Phi 300 \times 240$ mm, which can be used to determine the shear strength of coarse-grained soil with grain sizes below 60 mm. The vertical and horizontal loads are applied by a ball screw servo motor, with an output load of up to 300 kN. Soil sample deformation can be determined using a displacement sensor, and the shear rate of the testing machine can be controlled from 0.01 to 3.0 mm/min. In the testing machine, the upper and lower shear boxes and ball screw ends were connected by pins, allowing forward and reverse reciprocating shears.

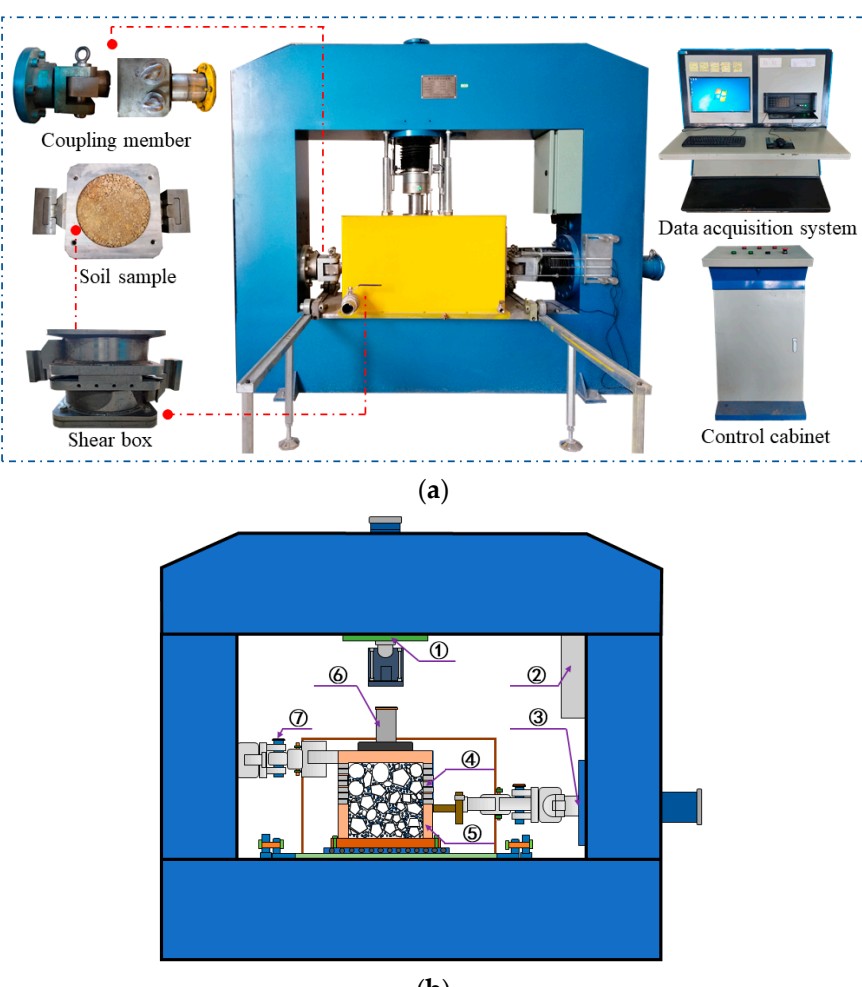

**Figure 1.** DHJ-30 coarse-grained soil shear testing machine. (**a**) DHJ-30 coarse-grained soil shear testing machine; (**b**) testing machine structure schematic. ① Vertical servo loading mechanism; data acquisition device; ③ ball screw servo loading motor drive system; ④ upper shear box; ⑤ lower shear box; ⑥ force transmission column; ⑦ pin.

### 2.2. Test Materials

The soil materials for the tests were collected from the 1980 m dumping terrace of a phosphate mine, mainly composed of mudstone and sandstone bulk rocks with varying grain sizes. The soil was sampled along the 1980 m dumping terrace at different heights of the dump (the ratio of the sampling point's height, h, from the top of the dumping terrace

to the height, H, of the dumping terrace) and was recorded as sampling points 1 to 4 from top to bottom. Geotechnical screens were used to sieve the sampled soil materials, and the screen apertures were 2 mm, 5 mm, 10 mm, 20 mm, 40 mm, and 60 mm. The sieving results are shown in Table 1.

**Table 1.** Grain size distribution of argillaceous siltstone rockfill materials from the 1980 m dumping terrace.

| Sampling Points | Grain Size Component Mass Percentages/% | | | | | | | Coarse Grain Content/% |
| | <2 mm | 2~5 mm | 5~10 mm | 10~20 mm | 20~40 mm | 40~60 mm | >60 mm | |
| --- | --- | --- | --- | --- | --- | --- | --- | --- |
| Sampling point 1 | 42.25 | 30.19 | 4.92 | 3.58 | 4.06 | 4.88 | 10.12 | 27.56 |
| Sampling point 2 | 35.58 | 26.40 | 6.71 | 5.54 | 6.61 | 7.80 | 11.36 | 38.02 |
| Sampling point 3 | 22.53 | 16.13 | 9.41 | 8.89 | 11.27 | 12.45 | 19.32 | 61.34 |
| Sampling point 4 | 14.13 | 10.44 | 11.60 | 9.84 | 14.07 | 16.18 | 23.74 | 75.43 |

As shown in Table 1, the coarse grain content of the sampled soil material increases with the increase of the height of the dump, the large-size grains are mostly distributed at the bottom of the dumping terrace, and the small-size grains are mostly distributed at the top. According to [9], 5 mm was used as the dividing grain size of the coarse and fine grains in the soil, and the mass percentage of grains over 5 mm in size was used as the coarse grain content, $P_5$. The coarse-grained soil indoor shear testing machine allows the maximum particle size of the filled soil to be 60 mm. Therefore, in order to meet the needs of the laboratory test, the prototype grading of the soil materials is scaled according to [32]. The scale formula is as follows (1):

$$P_{5i} = \frac{p_5}{p_5 - p_0} p_{05i} \tag{1}$$

where $P_5$ is the coarse grain contents (%); $P_{5i}$ is the content of a particle group with a particle size greater than 5 mm after scaling (%); $P_{05i}$ is the content of a particle group with a particle size greater than 5 mm before scaling (%); $P_0$ is the percentage of the particle mass with a particle size greater than 60 mm.

The grain size grading of argillaceous siltstone rockfill materials was downscaled using Equation (1). The cumulative curves of the downscaled grain size grading are shown in Figure 2.

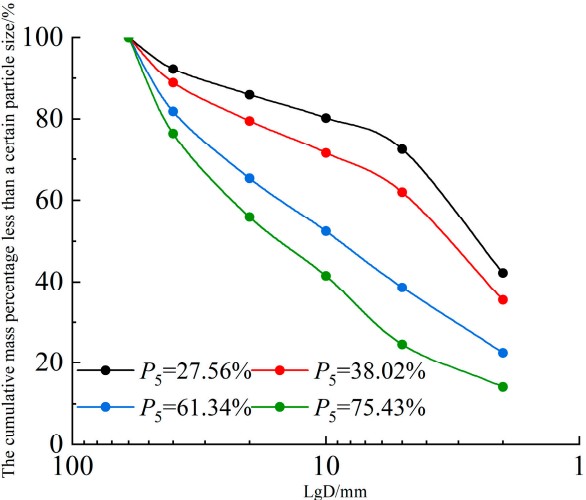

**Figure 2.** Particle size gradation accumulation curve.

### 2.3. Test Protocol

The tested soil materials had a natural moisture content of 3.04% and a natural bulk weight of 2.16 g/cm$^3$. Based on the soil grading composition in Table 1, four coarse grain contents of 27.56%, 38.02%, 61.34%, and 75.43% were designed. According to the physical property indexes of the tested soil materials, the dry soils required for samples with different coarse grain contents were weighed. The required amount of water was weighed and mixed well with the dry soils to prepare the remodeled soil samples with different coarse grain contents. The soil samples were loaded into test boxes in three layers and compacted. To avoid layering, the soil samples were chiseled after each compaction.

First, the ball screw servo motor system was used to push and shear the argillaceous siltstone rockfill materials with different coarse grain contents, and the soil samples were considered to fail as the shear strain reached 15% [33]. Then, the system was switched to pull and shear, which was stopped as the upper and lower shear boxes overlapped. The above process was repeated to achieve reciprocating push–pull shearing. The reciprocating shear paths are shown in Figure 3. $N = 1$ means one reciprocating push–pull shear. Soil samples in each coarse grain content group underwent four reciprocating push and pull shears, and the tests stopped. Thus, each soil sample underwent eight shears. During the reciprocating shear, the vertical compressive stress was set to 500 kPa, 1000 kPa, 1500 kPa, and 2000 kPa, and the horizontal shear rate was 1 mm/min. The test conditions were a solidified fast shear.

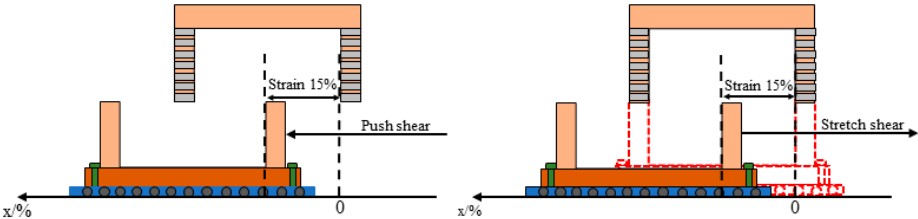

**Figure 3.** Reciprocating shear test path.

## 3. Analysis of Experimental Results

### 3.1. Setting of Relevant Mechanical Parameters

As the soil materials underwent reciprocating push and pull shears on the testing machine, hysteresis loops were formed on the shear stress–shear displacement curve [31], representing the energy loss from overcoming the frictional resistance of the soil grains during the reciprocating shear. The shear stress–shear displacement curves of coarse-grained soil under the reciprocating shear are shown in Figure 4. To facilitate the analysis, the A→B shearing process is denoted as a forward shear, and the shear strength is expressed as $\tau_T$; The B→A shear process is denoted as a reverse shear, and the shear strength is expressed as $-\tau_L$ (the negative sign denotes the direction); the area of the hysteresis loop is denoted by $S$.

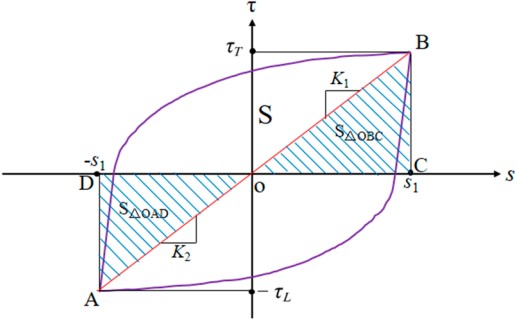

**Figure 4.** The schematic diagram of mechanical parameters related to reciprocating shear hysteresis loop of soil samples.

According to the relevant theories of geotechnics [31], the shear stiffness, *K*, and damping ratio, λ, of the tested argillaceous siltstone rockfill materials can be calculated using Equations (2) and (3).

$$K = \frac{K_1 + K_2}{2} \tag{2}$$

$$\lambda = \frac{\lambda_1 + \lambda_2}{2} = \frac{1}{2}\left(\frac{S}{4\pi S_{\Delta OBC}} + \frac{S}{4\pi S_{\Delta OAD}}\right) \tag{3}$$

where $K_1$ is the forward shear stiffness (kPa); $K_2$ is the reverse shear stiffness (kPa); $S$ is the hysteresis loop area (kPa·mm); λ is the damping ratio. $\Lambda_1$ is the damping ratio under the forward shear strength, and $\lambda_2$ is the damping ratio under the reverse shear strength.

### 3.2. Shear Stress–Shear Displacement Curve

Reciprocating shear tests were conducted on argillaceous siltstone rockfill materials with different coarse grain contents, and the test data were compiled according to Figure 4. Similarities were found in the development pattern of the shear stress–shear displacement curves of the argillaceous siltstone rockfill materials with different coarse grain contents under different vertical pressures. Using the data at 500 kPa as examples, the characteristics of the shear stress–shear displacement curves of the argillaceous siltstone rockfill materials with different coarse grain contents have been analyzed (Figure 5).

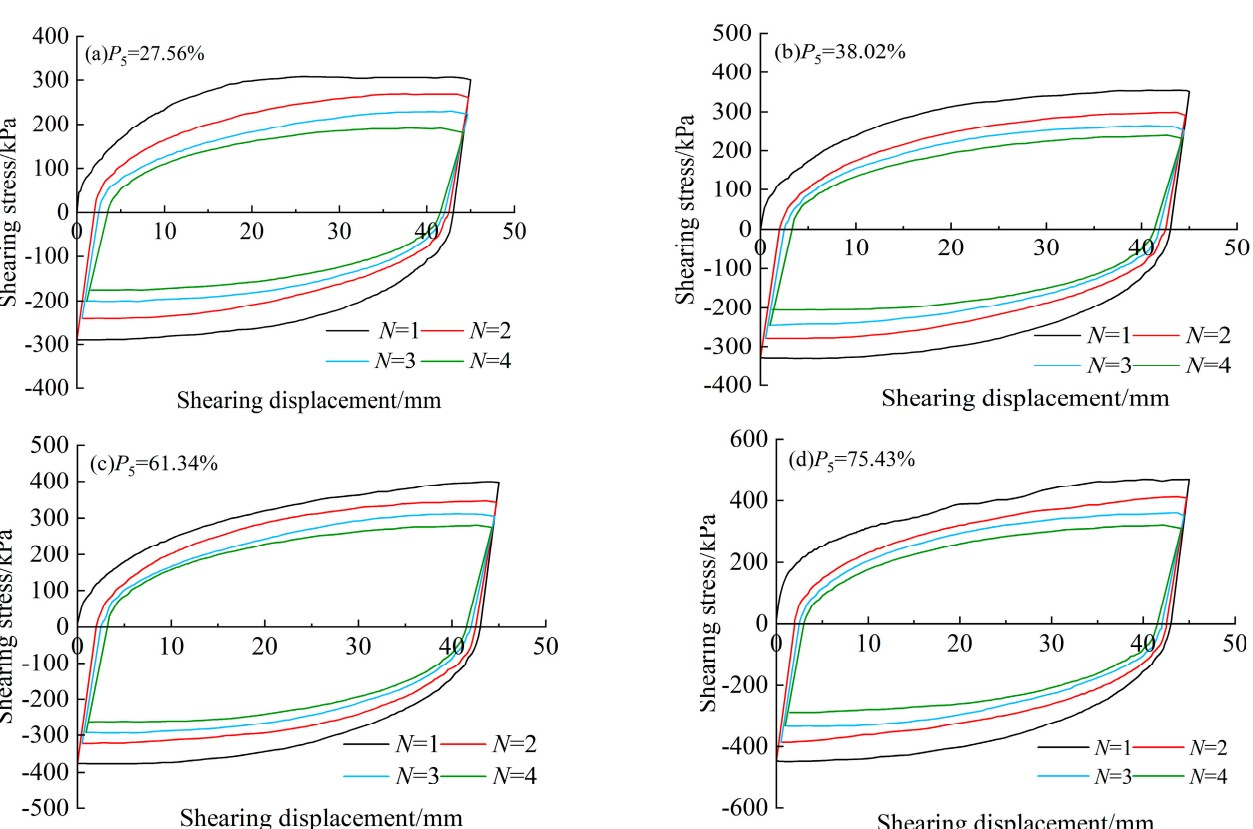

**Figure 5.** Reciprocating shear stress-shear displacement curves under different coarse particle contents.

As shown in Figure 5, the soils with different coarse grain contents under the reciprocating shear have strain-hardening-type stress–strain curves. The shear strength of the soil materials gradually increases with the coarse grain content. The reason is as follows. The skeleton is the main stress-bearing structure in the soils. At low contents, the coarse grains in the soil are in suspension, with insufficient interlocking among them. Thus, the ability to resist external force-induced deformation is relatively weak. With the gradual increase of coarse grains in the soil, the interlocking and contact among the block grains are

gradually enhanced. Since the soil skeleton is mainly formed by mutual contact among the coarse grains, the soil can resist greater shears, and the shear strength gradually increases. In addition, the shear strength of soil samples with the same coarse grain content varied in different reciprocating shear processes and decreased with the number of reciprocating shears, $N$. Specifically, when $N = 1$, the shear strength reduction of the soil samples is significant and greater than that during the other reciprocating shears. The reason is as follows. As the soil material undergoes the first reciprocating shear, the soil grains on the shear surface mutually compress, interlock, and closely contact each other, rendering it difficult for frictional sliding and tumbling of the soil grains. As the number of reciprocating shears increases, the grains on the shear surface are gradually rounded, and some grains fragment. As a result, the interlocking among the block grains decreases, and the friction characteristics weaken, gradually decreasing the shear strength of the soil material.

### 3.3. Shear Strength Analysis

By processing the data in Figure 5, we have the distribution of the forward shear strength, $\tau_T$, and reverse shear strength, $-\tau_L$, under 500 kPa of vertical pressure, as shown in Figure 6.

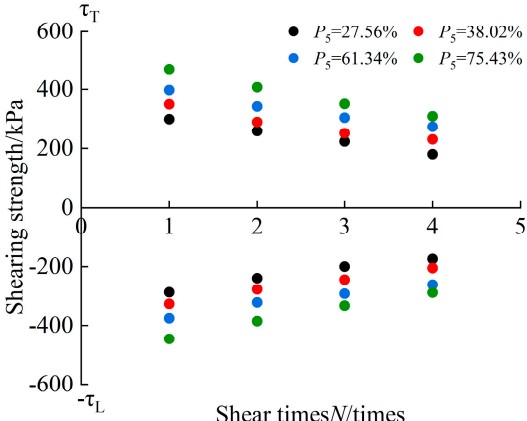

**Figure 6.** Reciprocating shear strength distribution map.

It can be observed that the shear strength of the argillaceous siltstone rockfill materials upon reciprocating shear failure is related to the coarse grain content and the number of reciprocating shears. With the same number of reciprocating shears, the shear strength of the soil samples increases gradually with the coarse grain content. At the same coarse grain content, the soil shear strength decreases with the number of reciprocating shears. In addition, with the same coarse grain content and the same number of shears, the forward shear strengths of the soil samples are greater than the reverse shear strengths, and the two are approximately symmetrically distributed about the x-axis.

### 3.4. Hysteresis Loop Area and Damping Ratio

After processing the test data based on Figure 4 and Equation (3), we have the hysteresis loop area, $S$, of the soil samples under 500 kPa of vertical pressure (Table 2). The distribution of the hysteresis loop area, $S$, and damping ratio, $\lambda$, are shown in Figures 7 and 8.

It can be observed from Table 2 and Figure 7 that under the same number of reciprocating shears, the hysteresis loop area of the soil samples is significantly affected by the soil fabric characteristics, and the hysteresis loop area of the soil samples increases as the coarse grain content increases. When the number of reciprocating shears is $N = 1$, for example, the hysteresis loop area, $S$, increases by 10.85%, 9.92%, and 19.73% as $P_5$ increases from 27.56% to 75.43%. The reason is that during the shearing process, the soil will produce energy loss due to overcoming the friction resistance, and the area of the hysteresis loop can be used to characterize the energy loss. As the number of coarse grains in the soil sample

increases, the contact and interlocking among the soil grains increases, and the external force required to induce shear deformation gradually increases. As a result, the energy loss of the soil sample to overcome the friction between the block grains also gradually increases, manifested as the increase of the hysteresis loop area.

**Table 2.** Grain size distribution of argillaceous siltstone rockfill materials from the 1980 m dumping terrace.

| Number of Shears $N$/Times | Hysteresis Loop Area $S$/kPa·mm | | | |
|---|---|---|---|---|
| | $P_5 = 27.56\%$ | $P_5 = 38.02\%$ | $P_5 = 61.34\%$ | $P_5 = 75.43\%$ |
| 1 | 21,651 | 24,000 | 26,380 | 31,586 |
| 2 | 16,583 | 18,537 | 21,885 | 24,934 |
| 3 | 13,628 | 16,022 | 19,055 | 21,881 |
| 4 | 11,431 | 13,815 | 17,003 | 18,912 |

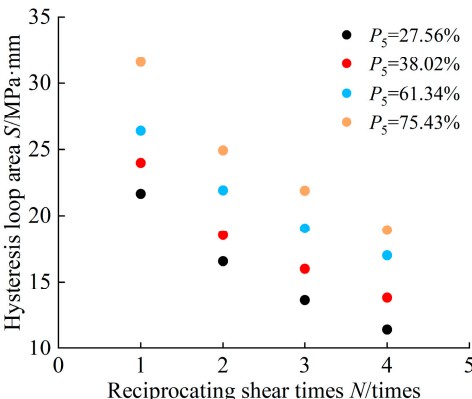

**Figure 7.** Hysteresis loop area distribution map.

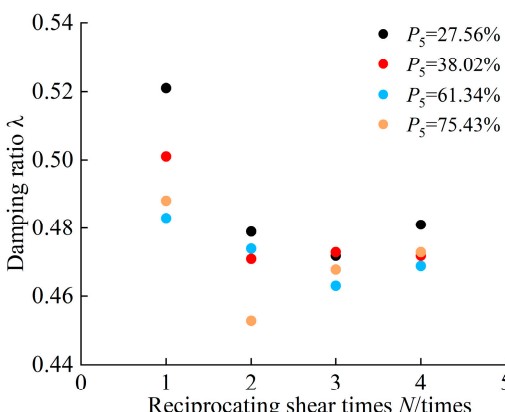

**Figure 8.** Damping ratio distribution diagram.

At the same coarse grain content, the hysteresis loop area of the soil samples gradually decreases with the number of reciprocating shears. Taking $P_5 = 27.56\%$ as an example, the decrease in the hysteresis loop area is 23.41%, 17.82%, and 16.12% when the number of reciprocal shears increases from $N = 1$ to $N = 4$. Among them, the decrease is larger from $N = 1$ to $N = 2$, and those during other shears are less significant. The possible reason is as follows. On the one hand, the contact and interlocking among the block grains on the shear surface are intense during the first reciprocating shear of the soil sample, and the frictional resistance is large. Therefore, the energy loss to overcome the frictional resistance is greater upon the shear failure of the soil sample, leading to larger hysteresis loop areas. On the other hand, as the number of reciprocating shears on the soil samples increases, the block

grains on the shear surface are relatively loose, due to the previous shear deformation, and gradually rounded. In the meantime, grain fragmentation reduces the grain size and weakens the interlocking, and the roughness of the shear surface decreases. In turn, the energy loss to overcome the frictional resistance of the soil grains upon shear failure of the soil sample is reduced. Therefore, the decrease in the hysteresis loop area is greater during the second reciprocating shear, and the decrease becomes slower with the number of reciprocating shears.

As shown in Figure 8, the damping ratios of the soil samples all tend to decrease with the number of reciprocating shears at the same coarse grain content. With the same number of reciprocating shears on the soil samples, the correlation between the damping ratio and coarse grain content is not significant, i.e., the damping ratio is not related to the coarse grain content.

### 3.5. Analysis of Shear Strength Parameter

As a common type of slope in geotechnical engineering, the shear strength parameter of dumping field slopes is an important index for analyzing and evaluating their stability. The experimental data are compiled according to Figure 6. The forward shear strength during each reciprocating shear is taken as the failure shear strength of that reciprocating shear, which is fitted with different normal stresses according to the Coulomb equation expressed in Equation (4). The fitting process is shown in Figure 9. The fitting results are shown in Table 3. How the cohesion and internal friction angle of the argillaceous siltstone rockfill material relate to the coarse grain content and the number of reciprocating shears is depicted in Figure 10.

$$\tau = \sigma \tan \varphi + c \tag{4}$$

where $\tau$ is the soil shear strength (kPa); $\sigma$ is the normal stress (kPa); $\varphi$ is the internal friction angle of the soil (°); $c$ is the soil cohesion (kPa).

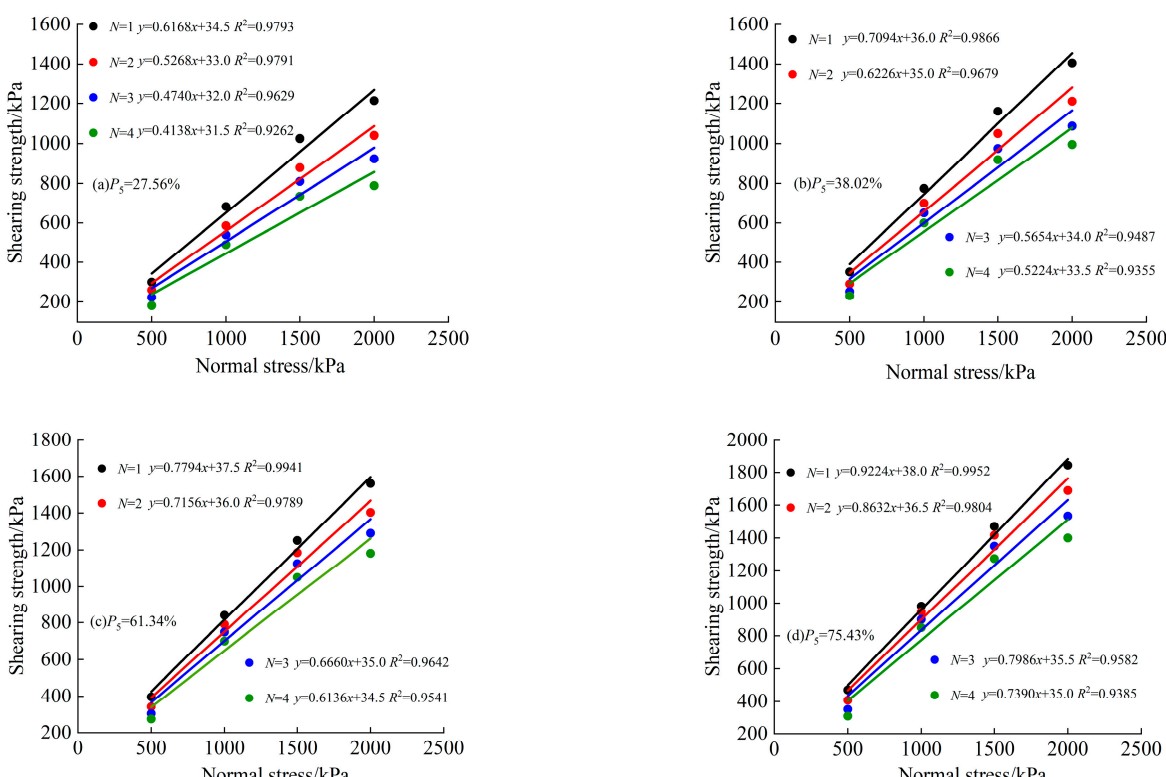

**Figure 9.** Rockfill shear strength parameter fitting diagram.

**Table 3.** Summary of how the shear strength parameter relates to coarse grain content and the number of reciprocating shears.

| Coarse Grain Content $P_5$/% | Number of Reciprocating Shears $N$/Time | Cohesion $c$/kPa | Internal Friction Angle $\varphi$/° |
|---|---|---|---|
| 27.56 | 1 | 34.50 | 31.68 |
| | 2 | 33.00 | 27.79 |
| | 3 | 32.00 | 25.37 |
| | 4 | 31.50 | 22.49 |
| 38.02 | 1 | 36.00 | 35.37 |
| | 2 | 35.00 | 31.92 |
| | 3 | 34.00 | 29.50 |
| | 4 | 33.50 | 27.60 |
| 61.34 | 1 | 37.50 | 37.95 |
| | 2 | 36.00 | 35.61 |
| | 3 | 35.00 | 33.68 |
| | 4 | 34.50 | 31.55 |
| 75.43 | 1 | 38.00 | 42.64 |
| | 2 | 36.50 | 40.80 |
| | 3 | 35.50 | 38.63 |
| | 4 | 35.00 | 36.48 |

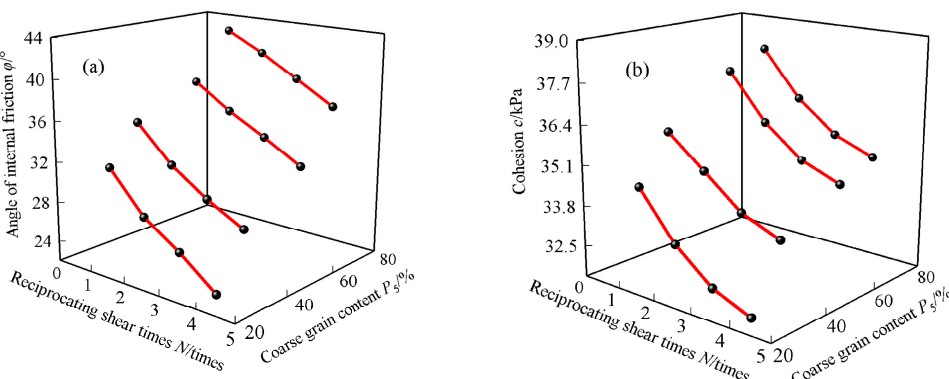

**Figure 10.** The relationship of shear strength parameters with coarse grain content and shear times. (**a**) Internal friction angle; (**b**) cohesion.

As shown in Figure 10, the cohesion and internal friction angle of the argillaceous siltstone rockfill material are positively correlated with the coarse grain content during the reciprocating shear, and they both gradually increase with the coarse grain content. Taking the first reciprocating shear as an example, the cohesion increases by 10.14% from 34.5 kPa to 38 kPa as the coarse grain content increases from $P_5 = 27.56\%$ to $P_5 = 75.43\%$, and the internal friction angle increases by 34.60% from 31.68° to 42.64°. The main reason is as follows. At low coarse grain contents, the contact and interlocking among the soil grains of the rockfill material are not sufficient, and the soil grains are in suspension, with weak interlocking. With the increase of the coarse grain content, the block grains increase. The coarse–fine grain contact and interlocking are sufficient and stronger, and the shear stress required for soil shear failure gradually increases. Accordingly, the cohesion and internal friction angle gradually increase.

Figure 10 also shows that the cohesion and internal friction angle of the argillaceous siltstone rockfill material are negatively correlated with the number of reciprocating shears, and they both gradually decrease with the number of reciprocating shears. Taking the coarse grain content $P_5 = 27.56\%$ as an example, the cohesion decreases by 4.35%, 3.03%, and 1.56% with the number of reciprocating shears, and the internal friction angle decreases by 12.28%, 8.71%, and 11.35%, respectively. The reason is as follows. During the first reciprocating

shear, the contact among the soil grains is close, with sufficient interlocking, leading to the greater shear stress required for soil shear failure. Accordingly, the cohesion and internal friction angle are greater. As the number of reciprocating shears increases, the spatial structure of the soil grains on the shear surface is disturbed and destroyed. The soil grains, with close contact and interlocking, leave their original positions through rotating, tumbling, and lifting by the reciprocating shear. As a result, the shear strength decreases, which ultimately leads to decreases in the cohesion and internal friction angle. As the reciprocating shear proceeds to a certain degree, the soil spatial structure is in equilibrium, and the cohesion and internal friction angle tend to stabilize.

## 4. Mechanical Property Evolution Analysis

### 4.1. Failure Shear Stiffness $K_f$

Shear stiffness is an important parameter characterizing soil shear deformation properties. According to [33], the shear stiffness corresponding to 15% of the strain in the soil samples can be defined as the failure shear stiffness. In this study, the failure shear stiffness during the first push shear on the soil samples with different coarse grain contents (Figure 5) is considered the failure shear stiffness of each coarse grain content group, denoted by $K_f$. The test results are summarized in Table 4.

**Table 4.** Failure shear stiffness, $K_f$, of argillaceous siltstone rockfill materials with different coarse grain contents.

| Coarse Grain Content $P_5$/% | Shear Stress $\tau$/kPa | Strain $\varepsilon$/% | Failure Shear Stiffness $K_f$ (MPa/mm) |
|---|---|---|---|
| 27.56 | 300 | 15 | 2.000 |
| 38.02 | 351 | 15 | 2.340 |
| 61.34 | 398 | 15 | 2.653 |
| 75.43 | 470 | 15 | 3.133 |

The relationship between the failure shear stiffness, $K_f$, and coarse particle content, $P_5$, was fitted and analyzed, and the results are shown in Table 5.

**Table 5.** The fitting relationship between failure shear stiffness, $K_f$, and coarse particle content, $P_5$.

| Model | Fitting Model Expression | Degree of Freedom | Residual Sum of Squares | Mean Square Deviation | Fitting Coefficient |
|---|---|---|---|---|---|
| Exponential model | $K_f = 1.6165 e^{0.0086 P_5}$ | 2 | $2.51 \times 10^{-2}$ | 13.15 | 0.9343 |
| Linear model | $K_f = 0.0217 P_5 + 1.4316$ | 2 | $2.40 \times 10^{-2}$ | 0.67 | 0.9482 |
| Logarithmic model | $K_f = 1.0267 \ln N - 1.4196$ | 1 | $2.45 \times 10^{-2}$ | 0.34 | 0.9462 |
| Power model | $K_f = 0.5022 P_5^{0.4168}$ | 2 | $3.01 \times 10^{-2}$ | 13.15 | 0.9250 |

It can be seen from Table 5 that the linear fitting of the relationship between the failure shear stiffness, $K_f$, and the coarse particle content, $P_5$, is the best, as shown in Figure 11.

According to Figure 11, the relationship between the failure shear stiffness, $K_f$, and the coarse grain content, $P_5$, of the soil samples can be expressed as follows:

$$K_f = 0.0217 P_5 + 1.4316 \qquad (5)$$

In Equation (5), the linear fit correlation coefficient is 0.9655, and the soil sample's failure shear stiffness, $K_f$, has a good linear correlation with the coarse grain content, $P_5$.

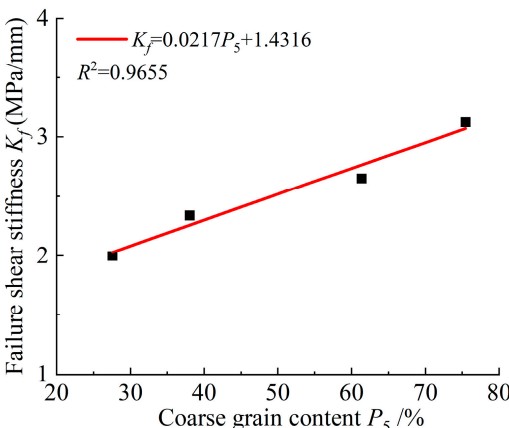

**Figure 11.** $K_f$-$P_5$ Relationship fitting curve.

### 4.2. Shear Stiffness, K

During the reciprocating shear tests, the soil sample's shear stiffness, $K$, can be calculated with Equation (2), where $K_1$ and $K_2$ are the forward failure shear stiffness and reverse failure shear stiffness, respectively. The variation of the shear stiffness ratio, $K/K_f$, with the number of reciprocating shears, $N$, can be obtained after eliminating the dimension of the soil's shear stiffness using the ratio method. At the same time, the relationship between the shear stiffness ratio, $K/K_f$, and the reciprocating shear times, $N$, was fitted and analyzed, and the results are shown in Table 6.

**Table 6.** The fitting relationship between failure shear stiffness ratio, $K/K_f$, and reciprocating shear times, $N$.

| Model | Fitting Model Expression | Degree of Freedom | Residual Sum of Squares | Mean Square Deviation | Fitting Coefficient |
|---|---|---|---|---|---|
| Exponential model | $K/K_f = 1.1201e^{-0.1449N}$ | 2 | $9.06 \times 10^{-4}$ | 1.28 | 0.9779 |
| Linear model | $K/K_f = -0.1133N + 1.0733$ | 2 | $6.46 \times 10^{-4}$ | 0.06 | 0.9851 |
| Logarithmic model | $K/K_f = -0.243 \ln N + 0.9834$ | 1 | $4.19 \times 10^{-6}$ | 0.03 | 0.9901 |
| Power model | $K/K_f = 0.9859N^{-0.2934}$ | 2 | $1.58 \times 10^{-3}$ | 1.28 | 0.9635 |

It can be seen from Table 6 that the logarithmic fitting of the relationship between the shear stiffness ratio, $K/K_f$, and the reciprocating shear times, $N$, is the best, as shown in Figure 12.

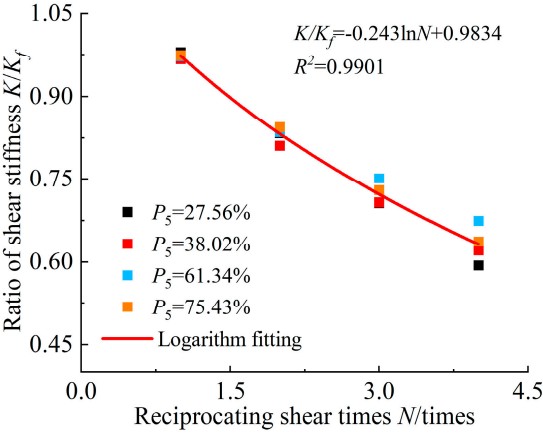

**Figure 12.** $K/K_f$-$N$ relationship fitting curve.

As shown in Figure 12, the $K/K_f$ has obvious nonlinear characteristics with the increase of the $N$. Fitting with the logarithmic function yields:

$$K/K_f = -0.243 \ln N + 0.9834 \tag{6}$$

Combining Equations (5) and (6), an empirical formula for the variation of the shear stiffness, K, with the coarse grain content and the number of reciprocating shears in the argillaceous siltstone rockfill materials can be established, as expressed in Equation (7).

$$K = (-0.0053P_5 - 0.3479) \ln N + 0.0213P_5 + 1.4078 \tag{7}$$

where $N$ is the number of reciprocating shears, $1 \leq N \leq 4$; $P_5$ is the coarse grain content, $0\% \leq P_5 \leq 75\%$.

The soil sample's shear stiffness, $K$, was calculated using Equation (7) and compared with the test values, as shown in Figure 13. The experimental error analysis is shown in Figure 14.

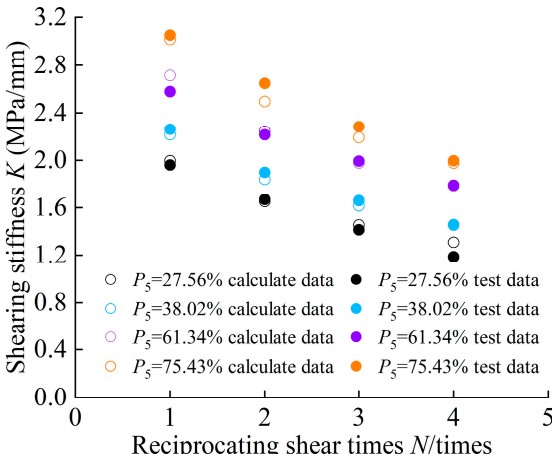

**Figure 13.** Comparison diagram between calculated value and test value.

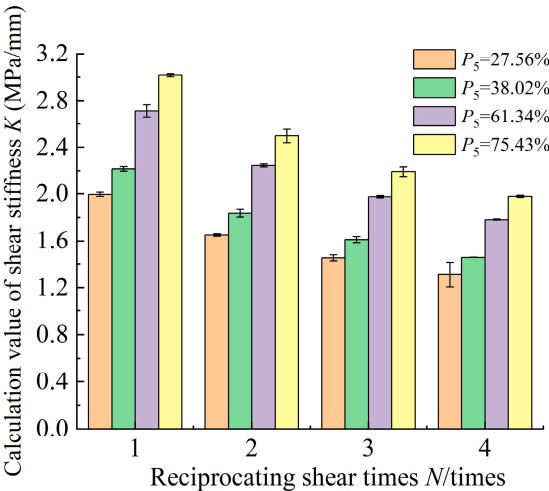

**Figure 14.** Error analysis chart.

According to Figures 13 and 14, the calculated shear stiffness of the soil sample has a small error with the test value, indicating that Equation (7) can be used to characterize the variation of the shear stiffness of argillaceous siltstone rockfill material with the number of reciprocating shears and coarse grain contents.

## 5. Strength Deterioration Mechanism Analysis and Discussion

Rockfill material is a dispersed system composed of soil particles with different particle sizes and shapes. Under the action of compaction, the coarse particles contact each other to form the soil skeleton, and the fine particles fill the pores of the coarse particles with cementation. The close contact structure of the coarse and fine particles gives the soil the ability to resist shear deformation. The shear strength of the rockfill material is composed of the apparent cohesion and friction strength of the soil [34]. The apparent cohesion reflects the adhesion degree of fine particles in the soil and the interlocking strength of coarse particles. The friction strength is a macroscopic representation of the frictional interaction of soil particles on the shear surface and the degree of directional particle arrangement.

Figure 15 shows the shear surface characteristics of a soil sample with 61.34% coarse grain content after the reciprocating shear under the normal stress of 500 kPa. It can be seen that under the reciprocating shear, the block particles distributed on the shear surface of the soil crush, the effect of the squeezing occlusion between the coarse particles weakens, and the directionally arranged fine particles on the shear surface significantly increases, resulting in the continuous deterioration of the soil shear strength.

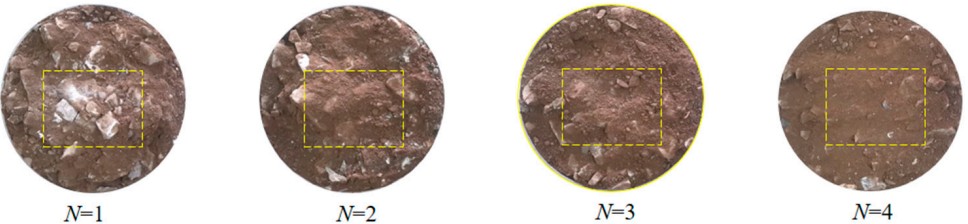

**Figure 15.** Shear surface characteristics of soil sample after reciprocating shear.

The meso-mechanism of the shear strength deterioration of rockfill materials under reciprocating shear is shown in Figure 16.

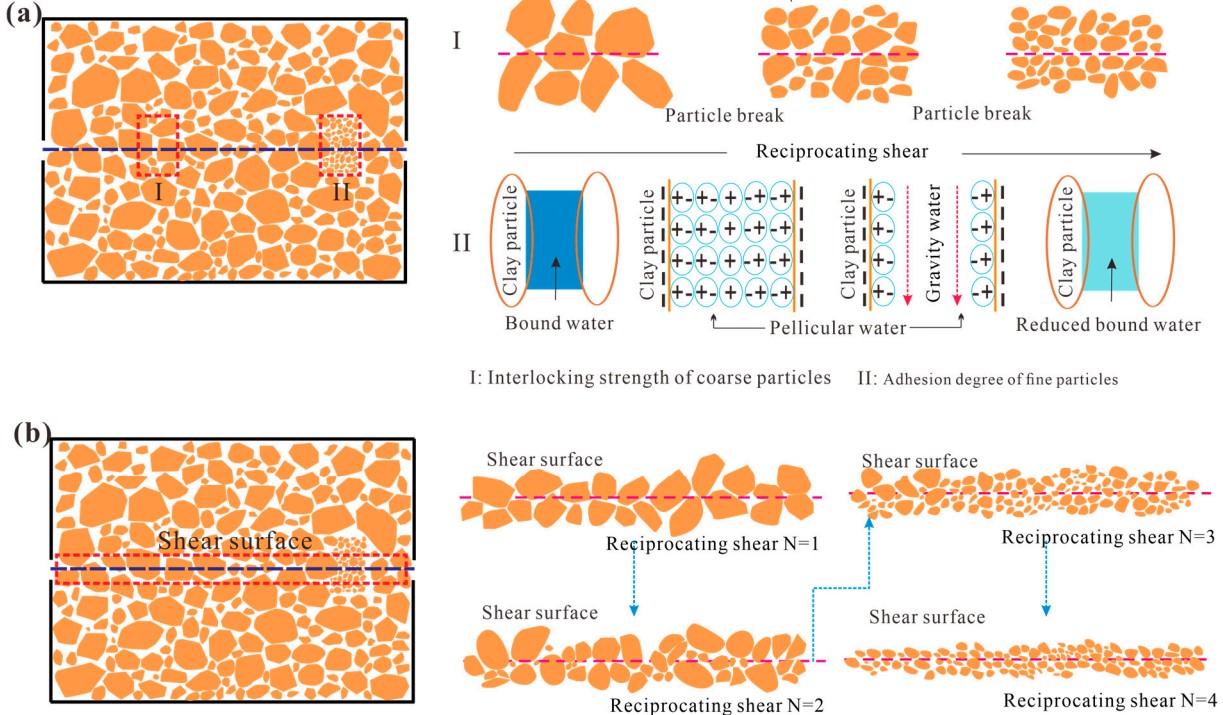

**Figure 16.** Schematic diagram of deterioration mechanism of shear strength of rockfill materials under reciprocating shear. (**a**) Deterioration of apparent cohesion. (**b**) Deterioration of friction strength.

Figure 16a illustrates the deterioration of the apparent cohesion of the rockfill material under reciprocating shear. A change in the original spatial distribution and contact mode of the soil particles when the shear stress is generated in the soil can be seen. In order to obtain a larger shear space, the soil particles will squeeze, roll, and rotate to change the position, thus forming a shear surface in the soil. However, with the continuous increase of shear stress in the soil, the particle occlusion is closer, the stress concentration is more obvious, and the number of cracks inside the block stone increases. Furthermore, the reciprocating shear accelerates the particle breakage, with an increasing number of internal fractures. In the alternating changes of push and pull environments, these fractures gradually open, widen, deepen, and penetrate. As a result, the disintegration of the block particles continuously occurs, and the particle edges are gradually smoothed, with increased roundness. The interlocking effect between the particles continuously weakens, and the apparent cohesion of the soil body further deteriorates. In addition, the fine particles in the rockfill material have good water-holding properties. The fine particles are dissociated and negatively charged by water. After forming an electric field, the cations in the aqueous solution are tightly adsorbed on the surface of the fine particles through electric field attraction to form a bound water membrane, exerting a certain bonding effect between the fine particles. With the increase of the number of reciprocating shears, the compactness of the shear surface of the soil decreases, the pores between the soil particles increase and connect with each other, and the electric field attraction is insufficient to overcome the gravitational water flow, resulting in the decrease of the binding strength between the fine particles and the decrease of the apparent cohesion of the soil.

Figure 16b illustrates the particle distribution characteristics on the shear surface of the rockfill material under reciprocating shear. It can be seen that under reciprocating shear, the block particles distributed on the shear surface of the soil are broken, the number of fine particles significantly increases, the roundness of the soil particles increases, and the interlocking friction between the soil particles decreases significantly due to the decrease in particle size. In addition, after repeated shear deformation, the compactness of the shear surface is weakened, and the frictional impedance effect of the soil is weakened by the increasing number of fine particles directionally arranged on the shear surface.

Although the contact mode and directional arrangement of the soil particles will continue to change during the process of reciprocating shear deformation, the interlocking and friction impedance effects between the soil particles will deteriorate. However, with the soil subjected to repeated reciprocating shear, the shear surface of the soil gradually penetrates and is in a loose state, and the ability of the soil to resist shear failure tends to be stable. Therefore, the complex physical–mechanical behavior response of argillaceous siltstone rockfill materials during reciprocating shear causes them to exhibit significant strength deterioration characteristics. Both forward and reverse shear strengths decrease with the number of reciprocating shears. The development and evolution of the soil sample's hysteresis loop area, damping ratio, and shear stiffness are similar to those of its shear strength, all exhibiting different degrees of deterioration with the number of reciprocating shears.

## 6. Conclusions

(1) The shear strength of argillaceous siltstone rockfill materials is closely related to and significantly affected by the coarse grain content and the number of reciprocating shears. The shear strength increases with the coarse grain content and decreases with the number of reciprocating shears. In addition, the forward shear strength is greater than the reverse shear strength, and the two are approximately symmetrically distributed about the x-axis.

(2) The hysteresis loop area and damping ratio of soil samples with the same coarse grain content tend to decrease with the number of reciprocating shears. With the same number of reciprocating shears, the hysteresis loop area of the soil samples increases

gradually with the coarse grain content, while the damping ratio is not correlated with the coarse grain content.

(3) The shear strength parameter of argillaceous siltstone rockfill materials is significantly affected by the coarse grain content and the number of reciprocating shears. The internal friction angle and cohesion increase with the coarse grain content and gradually decrease with the number of reciprocating shears.

(4) When the coarse grain content is $P_5$ = 27.56%, $P_5$ = 38.02%, $P_5$ = 61.34%, and $P_5$ = 75.43%, respectively, the shear stiffness of argillaceous siltstone rockfill increases linearly with the increase of the coarse grain content. The shear stiffness ratio has a logarithmic function relationship with the number of reciprocating shears. The shear stiffness of argillaceous siltstone rockfill is related to the coarse grain content and the number of reciprocating shears.

(5) The variability of the mineral composition and fabric characteristics of rockfill materials is the principal cause of their strength deterioration, and reciprocating shear serves as the trigger and changes the original spatial soil grain distribution and contact pattern. As the grain fragmentation intensifies, the interlocking among the block grains weakens, and the soil shear strength shows a trend of gradual deterioration.

**Author Contributions:** Conceptualization, J.D. and Z.X.; methodology, D.L., X.S. and W.Z.; validation, D.L., C.L. and W.Z.; resources, J.D.; data curation, Z.X.; writing—original draft preparation, J.D., Z.X. and D.L.; writing—review and editing, X.S., C.L. and W.Z.; supervision, J.D.; project administration, J.D.; funding acquisition, J.D. All authors have read and agreed to the published version of the manuscript.

**Funding:** This work is supported by the Special Basic Cooperative Research Programs of the Yunnan Provincial Undergraduate University's Association (202101BA070001-137), the Basic Research Project of the Yunnan Province of China (202101AT070144), and the Talent Introduction Program of Kunming University (XJ20220015).

**Data Availability Statement:** The original contributions presented in the study are included in the article, and further inquiries can be directed to the corresponding author.

**Acknowledgments:** The authors particularly appreciate the valuable comments made by the editors and reviewers to make a substantial improvement to this manuscript.

**Conflicts of Interest:** The authors declare that the research was conducted in the absence of any commercial or financial relationships that could be construed as a potential conflicts of interest.

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
