# Peer review of "Experimental Study on the Reciprocating Shear Characteristics and Strength Deterioration of Argillaceous Siltstone Rockfill Materials"

_applsci, doi:10.3390/app13158888_

Round 1
Reviewer 1 Report
In this work, the authors studied the mechanical properties of coarse-grained soils using a direct shear testing machine. Shear strength, hysteresis loop area, and damping ratios showed differences at various grain contents, attributed to the changes in internal friction angle and cohesion. However, the results need more consolidation with more experiment conditions and repeated tests (missing error bars). This work requires at least a major revision before further consideration in Applied Sciences.
1. The authors claim that grain fragmentation is one of the main mechanisms for stress-shear displacement and hysteresis loop area differences in different reciprocating numbers. Can the author comment on the possibility of evaluating the degree of grain fragmentation? For example, how to evaluate the change of grain size/shape before and after shear tests?
2. In Figure 8, the P5 sample shows a significant drop in the damping ratio in N = 2, which is very different from all other groups. Is there any explanation?
3. What is the repeatability of all experiments and the confidence of the results? There is no error bar in all data.
4. In Figure 13, the authors introduced the concept of the “Shear plane”. However, there is no relevant information in the main content.
5. What is the implication of this work to applications?
English grammar should be improved. For example, on Page 9 Line 3, “Since the hysteresis loop area represents the energy loss from overcoming the frictional resistance of the soil grains during the reciprocating shear.”
Author Response
Thanks to the reviewer for your valuable comments. The author revised the manuscript according to the reviewer's comments, and the revisions are marked in red in the manuscript. Then answer the questions raised by the reviewer one by one.
Please see the attachment

Reviewer 2 Report
I find the manuscript well written and properly organized. However, I have a number of issues that should be addressed in the resubmitted version of the paper. The list is presented below.
- There are no line numbers, which makes the review process less professional and more problematic.
- What is the threshold value used for the coarse-grained soil? Provide this infomation before the description of Equation (1)?
- What is the measurement frequency of the laboratory machine used?
- Have you considered digital image correlation (DIC) possible application?
- Please, show at least one real image of the analyzed samples.
- Chapter 2.2 - what are the precise locations (h/H) of the sampling points? Is it just one sample per specific location?
- Figure 2 - clarify in the text that this is a semi-logarithmic scale.
- Chapter 2.3, line 1- was the moisture and density the same for every sampling point indeed? Or is it an averaged value?
- Figure 4 - it is suggested by the Figure that K_1 and K_2 are equal.
- Text below Figure 4 - use the subscripts consequently in the text and in the figure (eg. \tau_T).
- Page 6- "According to the relevant theories of geotechnics" - references herein are necessary.
- Equation (3) - explain \lambda_1 and \lambda_2.
- Figure 6 - place the legend on the right of the plot. In the present form, it makes the Figure a bit messy.
- Chapter 3.5 - "The fitting results are shown in Table 3" - Describe the fitting procedure in a more detailed way.
- Equation (5) - have you studied other than linear fit?
- Equation (6) - have you studied other than logarithmic fit?
- Equation (7) is based on the two previously defined Equations (5) and (6). However, there is no background presented for those Equations. Probably, only the numerical evidence can be shown. In such a case, it suggested to use various fitting functions for Equations (5) and (6) in order to demonstrate the superiority of the specific combination. Present the corresponding error analysis.
- What do you mean by "fabric"? And consequently "fabric characteristics"?
- Conclusion (4) - Please, rephrase this conclusion. Firstly, it just a specific behavior observed for the analyzed samples and it cannot be generalized. Secondly, the character of this behavior was only assumed by the Authors.
Author Response

(The authors gave the same response as above.)

Round 2
Reviewer 1 Report
All comments have been addressed